# Agarose-Based Hydrogel Film with Embedded Oriented Photonic Nanochains for Sensing pH

**DOI:** 10.3390/polym16111530

**Published:** 2024-05-29

**Authors:** Dunyi Xiao, Huiru Ma, Wei Luo, Jianguo Guan

**Affiliations:** 1School of Materials Science and Engineering, Wuhan University of Technology, Wuhan 430070, China; 317288@whut.edu.cn; 2School of Chemistry, Chemical Engineering and Life Science, Wuhan University of Technology, Wuhan 430070, China; 3State Key Laboratory of Advanced Technology for Materials Synthesis and Processing, International School of Materials Science and Engineering, Wuhan University of Technology, Wuhan 430070, China; guanjg@whut.edu.cn; 4Wuhan Institute of Photochemistry and Technology, 7 North Bingang Road, Wuhan 430083, China

**Keywords:** responsive photonic nanochains, pH-responsive photonic crystals, fast response, photonic crystal heterogeneous gels

## Abstract

Responsive photonic crystal hydrogel sensors are renowned for their colorimetric sensing ability and can be utilized in many fields such as medical diagnosis, environmental detection, food safety, and industrial production. Previously, our group invented responsive photonic nanochains (RPNCs), which improve the response speed of photonic crystal hydrogel sensors by at least 2 to 3 orders of magnitude. However, RPNCs are dispersed in a liquid medium, which needs a magnetic field to orient them for the generation of structural colors. In addition, during repeated use, the process of cleaning and redispersing can cause entanglement, breakage, and a loss of RPNCs, resulting in poor stability. Moreover, when mixing with the samples in liquid, the RPNCs may lead to the contamination of the samples being tested. In this paper, we incorporate one-dimensional oriented RPNCs with agarose gel film to prepare heterogeneous hydrogel films. Thanks to the non-responsive and porous nature of the agarose gel, the protons diffuse freely in the gel, which facilitates the fast response of the RPNCs. Furthermore, the “frozen” RPNCs in agarose gel not only enable the display of structural colors without the need for a magnet but also improve the cycling stability and long-term durability of the sensor, and will not contaminate the samples. This work paves the way for the application of photonic crystal sensors.

## 1. Introduction

Integrating stimulus-responsive hydrogels or polymers with photonic crystals allows the conversion of stimulus signals into visible color patterns [1]. Previous researchers have made substantial contributions in related studies, achieving ion responsiveness [2,3,4,5,6], molecular responsiveness [7,8,9,10,11,12,13], temperature responsiveness [14,15,16,17], and electrical responsiveness [18,19,20,21], playing a significant role in sensing, medical, environmental, and display fields. However, the use of traditional homogeneous hydrogels or polymers poses two challenges: first, hindered diffusion of analyte molecules through the hydrogel [22,23]; second, the relaxation time of the gel is directly proportional to the square of the gel size [24]. Both issues contribute to prolonged response times for responsive photonic crystals, thus affecting their practical applicability. Our team has previously reported the preparation of various responsive photonic nanochains (PNCs) by using hydrogen-bond-guided template polymerization to orient steric-repulsion stabilized magnetic nanoparticles and lock them into one-dimensional structures [25,26,27]. The surface-responsive gel shell layer, only tens of nanometers thick, overcomes both issues, ensuring rapid response times for responsive photonic crystals in practical applications.

However, currently developed responsive PNCs, when utilized, can only be suspended in liquid, requiring a continuous application of the magnetic field to maintain the orientation for the revealing of structural colors. Additionally, during repeated use, the cleaning and redispersion process poses a risk of chain loss, entanglement, or breakage, leading to a deterioration in optical performance. Lastly, the suspended nature of the chain structure in the liquid makes it easy to cause sample contamination as it drifts with the solution.

To address the above issues, we propose the utilization of responsive PNCs as chromogenic units and their immobilization within a non-responsive porous network gel, resulting in the fabrication of heterogeneous gel films. The non-responsive porous gel chosen for immobilization of the PNCs is agarose gel, known for its tunable pore structure, with the diffusion rate of most analytes within agarose gel nearly equivalent to that in pure water [28,29]. This characteristic ensures the rapid response capability of responsive nanochains in agarose gel. Additionally, agarose is primarily derived from red algae, where it serves as the main polysaccharide component. Agarose gel exhibits excellent biocompatibility, making it commonly used in biomedical fields such as tissue engineering and drug delivery systems. The microstructure and composition of the film were characterized using scanning electron microscopy (SEM), optical microscopy, thermogravimetric analysis (Tg), and Fourier-transform infrared spectroscopy (FTIR). The optical and responsive properties of the heterogeneous film were investigated using a fiber optic spectrometer. Furthermore, the mechanical properties of the heterogeneous gel film were examined using a rotational rheometer.

## 2. Materials and Methods

### 2.1. Materials

Acrylic acid (AA, 99%), 2-hydroxethyl methacrylate (HEMA, 99%), 2-hydroxy-2-methylpropiophenone (HMPP, 97%), ethylene glycol dimethacrylate (EGDMA, 98%), 3-Acrylamide phenylboronic acid (AAPBA, 99%), N-(2-Hydroxyethyl) acrylamide (HEAAm, 98%), N,N′-methylenebis(2-propenamide) (BIS, 99%), Acrylic amide (AM, 99%), and agarose were purchased from Aladdin Reagent Co., Ltd. (Shanghai, China). Ethanol, dimethyl sulfoxide (DMSO), and ethylene glycol (EG) were purchased from Sinopharm Chemical Reagent Co., Ltd. (Shanghai, China). All chemicals were used directly as received without further purification. Deionized water (18.20 MΩ·cm) was produced in a Milli-Q system (Millipore, Burlington, MA, USA). Superparamagnetic Fe_3_O_4_@PVP nanoparticles were synthesized according to the previously reported method [30].

### 2.2. Preparation of pH RPNCs and Responsive Photonic Nanoparticles

pH RPNCs were synthesized according to the previously reported method [25]. First, we took 4 mL of ethanol dispersion containing Fe_3_O_4_@PVP nanoparticles, removed the supernatant after centrifugation, added 2 mL of EG, and sonicated until it was uniformly dispersed. The particle concentration of this solution was 15 mg/mL. A total of 720 μL of the above-mentioned dispersion was taken, into which 6.3 g of EG, 19.4 mg of the photoinitiator HMPP, 14.4 mg of the crosslinker EGDMA, 0.26 g of the monomer AA, 0.31 g of the monomer HEMA, and 2.04 g of deionized water were added. After ultrasonic mixing, a brown precursor solution was obtained, which was stored in the dark under refrigeration. After that, the above precursor solution was transferred into a 500 mL glass beaker, which was then placed in a uniform magnetic field of 180 Gs. The polymerization reaction was initiated by exposing the solution to UV light (365 nm wavelength, manufactured by Shenzhen Sankun Technology Co., Ltd. (Shenzhen, China)) for 5 min. Subsequently, RPNCs were washed several times with an ethanol–water mixed solution (50 vol%). The supernatant was decanted, and then 600 μL of the NaOH solution (ionic concentration of 50 mM) was added, mixed well, and left for magnetic settling. Finally, after decanting the supernatant, the RPNCs were dispersed in deionized water for magnetic settling and stored for later use.

### 2.3. Preparation of pH-Responsive Photonic Nanochains/Agarose Heterogeneous Gel Films

Firstly, 1 g of agarose was added to 10 mL of DMSO and heated to 80 °C with stirring for 1 h to completely dissolve the agarose, preparing a stock solution with a concentration of 0.1 g/mL. Then, 40 μL of the agarose containing the DMSO stock solution, 280 μL of deionized water dispersion containing pH RPNCs, and 90 μL of DMSO were mixed uniformly at 80 °C. Subsequently, adhesive pads with different thicknesses (having a circular cavity with a diameter of 20 mm in the middle) were attached to a glass substrate, and the glass substrates were heated to 60 °C. The above-mentioned mixture solution at 80 °C was transferred to the circular cavity, and a cover glass was placed on top to evenly fill the cavity with the mixture solution. A uniform magnetic field of 600 Gs was applied to orient the RPNCs along the magnetic field direction. Once the glass substrate cooled to room temperature, the pH-responsive heterogeneous hydrogel film was obtained. Finally, the pH-responsive heterogeneous hydrogel film was washed off the glass substrate with deionized water and rinsed three times with deionized water.

### 2.4. Characterizations

All digital photos in this paper were taken using an iPhone. Optical microscopy images were recorded using an optical microscope (Zeiss Axio Observer 5 M, Gottingen, Germany). The field emission SEM (FE-SEM) images were captured on a Hitachi S-4800 (Tokyo, Japan) scanning electron microscope. A NETZSCH-STA2500 (Yokohama, Japan) Regulus instrument was used to conduct thermal analysis in the air at room temperature to 1000 °C and a heating rate of 5 °C/min. A Nicolet 6700 (Massachusetts, America) spectrometer was used to collect Fourier transform infrared (FTIR) spectra in the range of 400–4000 cm^−1^ with a resolution of 4 cm^−1^. The reflectance spectra were all obtained by using a fiber optic spectrometer USB 2000+ (Shanghai, China). The shear stress–strain curve data were obtained using the Anton paar MCR 302 (Graz, Austria) rotational rheometer at room temperature with an angular frequency of 5 rad/s. Data processing and plotting were performed using Origin 2021, and the spectral analysis software used was BiaoQi SpecSuite 2018.

## 3. Results and Discussion

Figure 1 depicts the preparation process of the pH-responsive heterogeneous hydrogel film. Initially, a mixed solution containing pH RPNCs, agarose gel, and solvents was heated to 70 °C and then dropped into the cavity of the double-sided adhesive. Subsequently, it was covered with a glass cover (to ensure that the solution would not solidify too quickly, the glass substrate remained on the heating plate throughout the process). A uniform magnetic field was then applied to orient the RPNCs along the magnetic field lines, after which the heating was turned off and the temperature was allowed to decrease to room temperature. During cooling, the agarose molecules bound together via hydrogen bonding, intertwining to form an agarose gel network with a double helix structure, thereby immobilizing the oriented RPNCs. Due to the fixed particle spacing of the RPNCs, the structural color from PNCs remains stable during the solidifying process of agarose gel.

In the prepared pH-responsive heterogeneous hydrogel film, the agarose gel matrix itself is not pH-responsive. Changes in pH do not cause the agarose gel matrix to expand or contract, and its volume remains constant. On the other hand, after being immobilized within the agarose gel matrix, the RPNCs still exhibit pH responsiveness. Their chain length undergoes expansion or contraction with changes in pH, thereby altering the interparticle spacing and ultimately resulting in color changes. The agarose gel matrix serves merely as a scaffold in the system, maintaining the one-dimensional orientation of the pH RPNCs. The RPNCs, on the other hand, play a role in forming structural colors and responding to external stimuli. Specifically, when the pH-responsive heterogeneous hydrogel film transitions from a high-pH environment to a low-pH environment, or vice versa, the agarose gel matrix, unaffected by pH changes, remains unchanged in volume. However, the pH-responsive gel shell layers on the immobilized RPNCs are influenced by changes in the proton concentration, undergoing protonation or deprotonation processes. The transition between carboxylic acid and carboxylate groups leads to the contraction or expansion of the gel shell layer, ultimately resulting in changes in the interparticle spacing of the RPNCs, as described by the Bragg diffraction formula:(1)mλ=2ndsinθ
where *m* represents the diffraction order, *λ* is the wavelength of the diffracted light, *n* stands for the effective refractive index, *d* denotes the lattice spacing, and *θ* is the angle between the incident light and the lattice plane. Under the same testing conditions, for the same sample, the diffraction order, effective refractive index, and angle between the incident light and the lattice plane remain consistent. The wavelength of the diffracted light is only influenced by the lattice spacing, while changes in particle spacing result in changes in the diffracted wavelength. Macroscopically, this leads to observable changes in the overall color of the pH-responsive heterogeneous hydrogel film.

As shown in Figure 2a, the optical microscope and SEM images indicate that the product is composed of responsive chains and a gel network, with the RPNCs being oriented and fixed within the gel under a magnetic field. Optical microscopy reveals that the typical sample’s membrane thickness is around 100 μm, and the RPNCs in the solution are not freely floating and dispersed like those in a liquid. Instead, they are orderly arranged together, indicating the presence of a gel network structure that immobilizes them. The cross-sectional SEM images in Figure 2b also show that the RPNCs are attached to the walls of the macroporous structure, which may occur when the gel loses water and the chains adhere to the walls of the matrix gel pores. This observation indicates the presence of a gel structure, in addition to the chains, that forms aerogels after dehydration.

From the thermogravimetric curve shown in Figure 2c, it can be observed that during the process from 250 °C to around 400 °C, the sample undergoes the decomposition of organic components, resulting in a final residue content of 44.4% inorganic material. This result is compared with the thermogravimetric curve of the RPNCs (in Appendix A), where the final residue content of inorganic material in the PRNCs is 67.64%. With a chain concentration of 9 mg/mL in the pH-responsive heterogeneous gel film, the calculated inorganic content in the RPNCs is 6.084 mg/mL. With an agarose gel concentration of 7.5 mg/mL, the calculated final residue inorganic content matches the measured value at 44.16%. From the infrared spectrum of RPNCs in Figure 2d, the absorption band at 3416 cm^−1^ corresponds to the O-H stretching vibration peaks of HEMA and AA, while the peak at 1710 cm^−1^ represents the C=O stretching vibration peaks of carboxyl and ester groups. The absorption peaks at 1642 cm^−1^, 1444 cm^−1^, and 1292 cm^−1^ originate from the PVP shell layer of the particles, and the peak at 575 cm^−1^ arises from the stretching vibration of Fe-O. The smaller area and weaker organic absorption peaks indicate a lower content of the gel shell layer in the responsive photonic nanochains. In the infrared spectrum of agarose, the absorption band at 1647 cm^−1^ corresponds to the bending vibration peak of O-H, while the bands at 1075 cm^−1^ represent the stretching vibration peaks of C-O. The characteristic peaks at 930 cm^−1^ and 891 cm^−1^ are attributed to the characteristic absorption peaks of 3,6-anhydrogalactose and the C-H bending vibration absorption peaks of pyranose. The consistency of the infrared spectrum peaks among the pH-responsive heterogeneous hydrogel film, RPNCs, and pure agarose indicates that there are no new chemical bonds formed between PNCs and agarose gel, but rather a physical blend.

Figure 2e,f depicts the optical performance testing of the pH-responsive heterogeneous hydrogel film. The film exhibits excellent responsiveness in different pH environments. Typical samples were prepared and placed in different pH buffer solutions without applying a magnetic field, with the solution ion strength at 150 mM. The peak position of pH-responsive heterogeneous gel film shifted by nearly 110 nm, indicating a color change across the green–yellow–orange–red spectrum.

Achieving a rapid response to external analytes has always been a crucial process in advancing the practical application of responsive photonic crystal hydrogel materials. Figure 2g demonstrates that the pH-responsive heterogeneous gel film exhibits rapid responsiveness based on its unique structural advantages. During the process of changing the solution environment, the peak shift occurs very quickly. When typical samples are exchanged from a pH 4.5 to 8.0 buffer solution environment, the time taken to record the wavelength at which the peak reaches 90% equilibrium (referred to as *τ*_90_) is measured. The results indicate that the typical samples reach a stable *τ*_90_ time of approximately 9.3 s, which is several orders of magnitudes shorter compared to the time taken for other pH-responsive homogeneous films of similar thickness [5]. Although reducing the thickness of pH-responsive films to several micrometers may also accelerate the response time, they could not form a free-standing film due to the insufficient mechanical properties [31,32]. We also tested the response time curves of the pH-responsive heterogeneous gel film in different pH environments, as shown in Appendix A.

In addition, using a non-responsive gel to immobilize RPNCs brings another advantage, enhancing stability and convenience. Figure 2h demonstrates that over 20 cycles, the pH-responsive heterogeneous gel film can still maintain good responsiveness, with the peak position remaining fixed. Moreover, after being tested under the same conditions one month and two months later, it still exhibits good responsiveness. In comparison to the current development where the process of washing and redispersing RPNCs during repeated use can easily lead to losses, entanglement, or breakage, thereby deteriorating their optical performance, the introduction of agarose gel enables the pH-responsive heterogeneous gel film to overcome these issues. Photonic crystal heterogeneous gel films should be stored in a sealed container containing pure water to prevent evaporation. It is best to store them at room temperature (recommended temperature: 10–30 °C). Temperatures exceeding 50 °C may cause the non-responsive agarose matrix gel to undergo a solid–liquid transition. As shown in Appendix A, the storage modulus (G′) and loss modulus (G″) curves start to intersect when the temperature rises from room temperature to 68 °C, indicating that the gel film transitions from a solid state to a flowing sol state at this temperature. This behavior is related to the properties of the agarose used. We also tested the effect of different temperatures on the optical performance of the pH-responsive heterogeneous hydrogel film. In addition to examining the effect of temperature on the heterogenous gel film through rheological properties, we also tested its optical properties. During the temperature increase from 7 °C to 52 °C, no changes in optical performance were observed, particularly in the position of the diffraction peaks, as shown in Appendix A. Before use, the photonic crystal heterogeneous gel film can be rinsed 2–3 times with deionized water and then directly placed into the solution environment for testing.

Figure 3a–f depicts experimental investigations by immobilizing PNCs in agarose gels of varying concentrations and subjecting them to buffer solutions at different pH values. From Figure 3a–e, it can be inferred that the diffraction peak positions remain consistent in pH 8.0 buffer solutions, with only the peak width continuously increasing. This indicates that the portion of RPNCs embedded remains the same under different gel concentrations. However, during the process of immobilization, the ordered arrangement of RPNCs is slightly disturbed by the agarose gel matrix, leading to an increase in peak width.

In addition, the size of non-responsive gel pores also affects the response time of pH-responsive heterogeneous hydrogel films. The curve showing the peak position changes over time in Figure 3f provides a more intuitive demonstration of the effect of gel concentration on response time. At the same time point, hydrogel films with lower gel concentrations achieve equilibrium in a shorter time, with a shorter *τ*_90_. When transitioning the pH-responsive heterogeneous hydrogel film from a pH 8.0 buffer solution to a pH 4.5 buffer solution, hydrogen ions rapidly diffuse into the interior of the agarose gel network. However, in hydrogel films with higher gel concentrations, the pores are smaller, resulting in stronger confinement of the embedded RPNCs. Consequently, the peak shift in the diffraction of RPNCs is smaller upon pH changes. This phenomenon is more pronounced in pH-responsive heterogeneous hydrogel films with gel concentrations exceeding 10 mg/mL. There exists a significantly low reflectance peak around 620 nm, indicating the presence of particle spacing like that at pH 8.0.

Table 1 presents the comparison of diameters for pure gel films of different concentrations in buffer solutions with varying pH values. From the table, it can be observed that there is no significant volume change for the same agarose gel film in buffer solutions with different pH values (the same gel film was used for each gel concentration). Actual images can be found in Appendix A. Unlike responsive gel films, agarose gel, as a non-responsive gel material, does not undergo volume changes in different pH environments due to the absence of pH-responsive groups in its network structure. Combining the observed phenomenon, when RPNCs are embedded in agarose gel, a shift in diffraction peak occurs. It can be inferred that the only components undergoing deformation in the system are the RPNCs and most of the volume remains unchanged, thus avoiding the phenomenon reported in the literature where gel expansion leads to slower responsiveness.

Figure 4a,b demonstrates the diffraction spectra of pH-responsive heterogeneous hydrogel films formed by low-concentration gelatin gel-fixing responsive particles, which vary with the magnetic field. We synthesized responsive particles under conditions without applying a magnetic field (the preparation process is consistent with RPNCs, except that no magnetic field was applied during polymerization) and used the same method with gelatin gel to prepare heterogeneous gel particle films under the same applied magnetic field. We found that under conditions without applying a magnetic field, structural colors could not be diffracted, indicating that RPNCs are indispensable in the formation of pH-responsive heterogeneous hydrogel films. Due to the lack of fixed-chain structures, responsive particles can move freely between pores, resulting in structural colors appearing with the application of a magnetic field. As the concentration of gelatin gel increases, the pore structure decreases, and the regular arrangement of particles under the magnetic field is restricted by spatial hindrance, leading to a decrease in reflectance. At a high gel concentration, as the pore structure begins to shrink, some of the particles’ orientation states can be fixed, allowing peaks to appear even without applying a magnetic field, as shown in Figure 4b. These observations are consistent with those reported by previous researchers [33].

We also investigated the effects of the chain concentration and film thickness on the optical, responsive, and mechanical properties of pH-responsive heterogeneous hydrogel films. Firstly, as shown in Figure 5a, it can be observed that with increasing chain concentration from 3 mg/mL to 9 mg/mL, the spectral reflectance increases by approximately 13%. This is because an increase in chain concentration leads to an increase in diffraction units within the pH-responsive heterogeneous hydrogel film. However, further increasing the chain concentration only results in slight fluctuations in reflectance, indicating that when RPNCs reach 9 mg/mL, the maximum limit of RPNCs that can diffract structural colors within the gel has been reached. During the incorporation of RPNCs, it was found that as the chain concentration increases, the pH-responsive heterogeneous hydrogel film becomes increasingly prone to damage. In Figure 5b, the shear stress–strain curves obtained under different chain concentrations by orienting RPNCs parallel to the direction of shear stress under a magnetic field show that pH-responsive heterogeneous hydrogel films with lower chain concentrations can withstand greater stress at the same strain, indicating that they are less likely to rupture during use. After considering the impact of the chain concentration on both optical and mechanical performance, we ultimately chose 9 mg/mL as the most suitable concentration for our needs.

Subsequently, an exploration of the influence of film thickness was conducted, with the chain concentration fixed at 9 mg/mL and the gel concentration at 7.5 mg/mL. As shown in Figure 5c, with increasing film thickness, the reflectance of the pH-responsive heterogeneous hydrogel film gradually increases until it reaches 200 μm, after which the rate of increase decreases. The increase in film thickness also represents an increase in the diffraction of structural color units. When the film thickness is small, the pH-responsive heterogeneous hydrogel film appears relatively transparent, allowing most of the light to pass through. As the film thickness increases, more RPNCs are present per unit area, leading to an increase in reflectance. However, the increase in film thickness leads to an increase in the distance for ion diffusion, requiring a longer time for ions to completely penetrate the pH-responsive heterogeneous hydrogel film and respond to the internal RPNCs. By recording the time required for the response to reach equilibrium when transferring pH-responsive heterogeneous hydrogel films of different thicknesses from a pH 8.0 to a pH 4.5 buffer solution, it was found that thicker films require a longer equilibrium time. In Figure 5d, the 50 μm thick film with a chain concentration of 9 mg/mL could not withstand the replacement of the buffer solution due to its low mechanical strength, resulting in fracture, and only the response time data for the 75 μm film were obtained. From the perspective of response time, a film thickness of 100 μm is required to achieve equilibrium within 30 s.

We further tested the mechanical properties of pH-responsive heterogeneous gel films with different thicknesses. The shear stress–strain curves are shown in Appendix A. It is evident that the mechanical performance significantly improves with increasing film thickness, which is consistent with the observations from our experiments.

The difference in trends between 8.0 to 4.5 and 4.5 to 8.0 shown in Figure 5d may be due to the fact that the responsive chains, when undergoing responsive deformation, induce an elastic deformation in the surrounding agarose gel. When the heterogeneous gel film is in the 8.0 buffer solution, the interparticle spacing within the chains is closer to the fixed state. Thus, during the response process from 8.0 to 4.5, the responsive chains need to overcome a certain amount of elastic deformation, increasing the response time. Conversely, in the transition from 4.5 to 8.0, this elastic deformation aids the responsive chains in quickly returning to their initial state.

## 4. Conclusions

In this study, RPNCs were utilized as both responsive and chromogenic units, while non-responsive gels with adjustable pore sizes served as matrices for immobilizing the responsive materials. Through chemical or physical fixation, two types of PNCs with distinct responsive properties were successfully anchored post-magnetic orientation, leading to the fabrication of fast-response heterogeneous gel films embedding one-dimensional oriented photonic PNCs. The mechanism underlying the fixation of RPNCs in low-concentration gels was elucidated, and the responsive performance, mechanical properties, and stability of photonic crystal heterogeneous gel films were thoroughly investigated. Compared to materials prepared using traditional homogeneous gel films, those fabricated with heterogeneous gel films exhibit the advantage of faster response speed. Furthermore, unlike the RPNCs previously developed by our group, the heterogeneous gel films do not necessitate the application of a magnetic field to display structural color during use. They also demonstrate significant improvements in stability when placed and cycled repeatedly. Typically, a 100 μm pH-responsive heterogeneous gel film can achieve a color change in just 9.3 s as the pH of the environment changes from 4.5 to 8.0. When compared to their homogeneous counterparts with the same thickness, the response rate has been greatly improved. The peak shift of pH-responsive heterogeneous gel films exceeds 100 nm across the green–yellow–orange–red color spectrum when placed in different pH environments. After 20 cycles, the responsiveness of the heterogeneous gel films remained unchanged, showing consistent performance even after several months of intermittent cycling tests, demonstrating good cycle stability and long-term durability. The fast-response heterogeneous gel films embedding one-dimensional-oriented PNCs offer a novel structural model for developing rapid-response, high-performance photonic crystal films, providing a method for the preparation of subsequent photonic crystal heterogeneous gel detection materials.

## Figures and Tables

**Figure 1 polymers-16-01530-f001:**
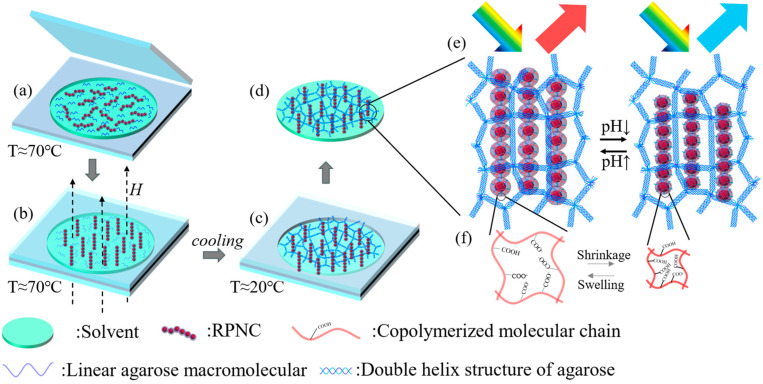
(**a**–**d**) are schematic of the preparation of pH-responsive heterogeneous gel film, the solution in the figure is a DMSO–water mixed solution (with DMSO 30 vol%), and the color shown in the figure is not the actual color of the solution; it is chosen merely for illustrative purposes. (**e**,**f**) are schematic diagrams illustrating the pH response mechanism.

**Figure 2 polymers-16-01530-f002:**
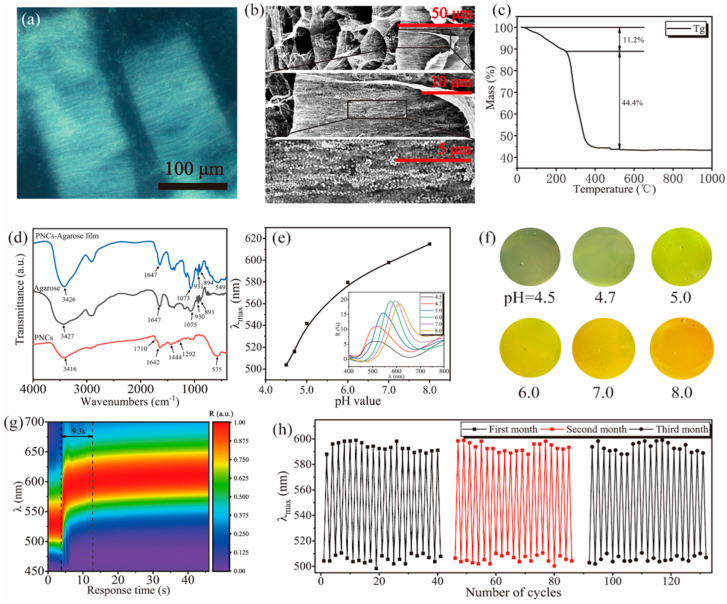
(**a**) Optical microscope images for cross–sectional view of oriented pH–responsive heterogeneous gel film; (**b**) SEM images and localized magnifications of the freeze–dried pH–responsive heterogeneous gel films; (**c**) TG curve of the dried pH–responsive heterogeneous gel film; (**d**) Infrared spectra of dried pH-responsive photonic nanochains, agarose gel film, and pH–responsive heterogeneous gel film; (**e**) diffraction peak position curves and diffraction spectra change with pH values in different buffer solutions for the pH–responsive heterogeneous gel film; (**f**) photographs for pH–responsive heterogeneous gel film in buffer solutions with different pH values; (**g**) the diffraction spectra of the 100 μm thick pH–responsive heterogeneous gel film change over time as it transitions from pH 4.5 to 8.0 buffer solution; (**h**) stability tests for the pH–responsive heterogeneous gel film in pH 4.5 and 8.0 buffer solutions; stability tests were conducted after one month and two months of placement.

**Figure 3 polymers-16-01530-f003:**
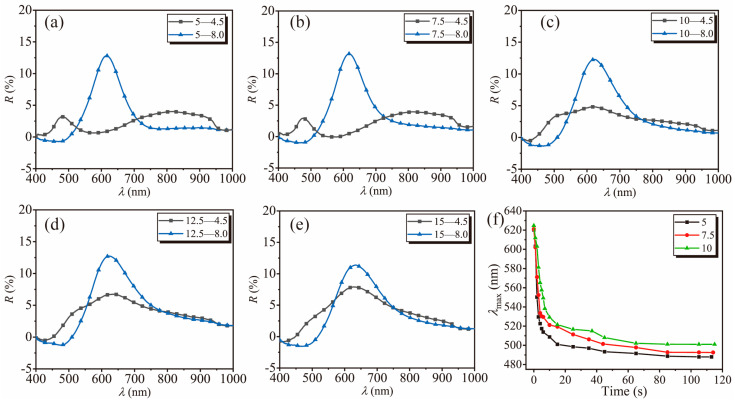
(**a**–**e**) Digital diffraction spectra images for pH−responsive HGF in buffer solutions with pH 4.5 and 8.0, at different agarose gel concentrations. In the legend, the first digit (5–15) represents the agarose gel concentration in pH−responsive HGF (in mg/mL) and the second digit (4.5 and 8.0) represents the pH of the solution environment where pH−responsive HGF is present; (**f**) curve showing the variation (5, 7.5, and 10 mg/mL) of diffraction peak positions over time as pH−responsive HGF transitions from a buffer solution with pH 8.0 to pH 4.5.

**Figure 4 polymers-16-01530-f004:**
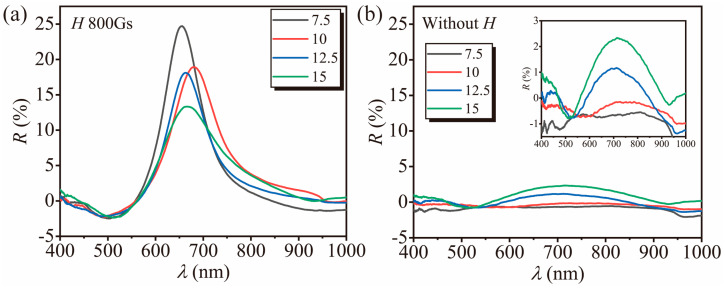
Diffraction spectra of heterogeneous gel films prepared with pH−responsive particles fixed in agarose gel at different concentrations (in mg/mL) under an applied magnetic field (**a**) and without (**b**). The illustration shows a modified version of (**b**) with an adjusted Y−axis to make the diffraction peaks more clearly visible.

**Figure 5 polymers-16-01530-f005:**
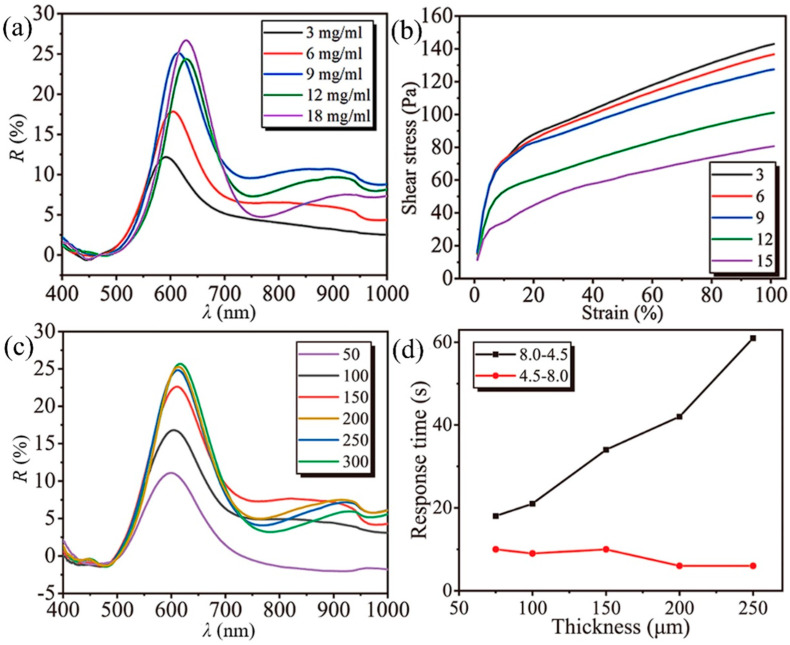
(**a**) Diffraction spectra for pH response of HGF with different chain concentrations, with a membrane thickness of 150 μm; (**b**) shear stress−strain curves for pH response of HGF with different chain concentrations; (**c**) diffraction spectra for pH response of HGF with different membrane thicknesses, with a chain concentration of 9 mg/mL; (**d**) response times for pH response of HGF with different membrane thicknesses in pH buffer solutions ranging from 8.0 to 4.5 (4.5 to 8.0).

**Table 1 polymers-16-01530-t001:** Table comparing the diameters measured for pure gel films of different agarose gel concentrations placed in pH 4.5 and pH 8.0 buffer solutions.

	Gel Concentration (mg/mL)	5	7.5	10
pH Value				
4.5		19.27	19.60	20.45
8.0		19.23	19.65	20.50

The unit for diameter in the table is mm.

## Data Availability

The original contributions presented in the study are included in the article/Appendix A, further inquiries can be directed to the corresponding authors.

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
