# Peer review of "Agarose-Based Hydrogel Film with Embedded Oriented Photonic Nanochains for Sensing pH"

_polymers, 2024, doi:10.3390/polym16111530_

Round 1
Reviewer 1 Report
Comments and Suggestions for Authors
The paper titled "Agarose-based hydrogel film with embedded oriented photonic nanochains for sensing pH" presents a new approach to developing responsive photonic crystal hydrogel films by incorporating one-dimensional oriented RPNCs within a non-responsive agarose gel matrix. While the research showcases originality and contributes to the field of responsive photonic crystal materials, the reviewer believes certain aspects could be further improved to enhance the work's comprehensiveness and practical relevance. The following suggestions are offered:
1. The paper mentions using UV light for initiating the polymerization reaction to form the RPNCs, but it does not provide details about the wavelength, intensity, or make of the UV light used. Please provide these details.
2. In Fig. 1, the solvent color does not match the description. Please clarify or correct this discrepancy.
3. Have you tested the effect of temperature changes on the hydrogel films?
4. It would be beneficial to discuss the potential environmental impact or biodegradability of the hydrogel films, particularly for applications in environmental sensing or biomedical fields.
5. How does the hydrogel film thickness affect the mechanical properties and flexibility of the material?
6. While the paper mentions the rapid response time of the hydrogel films, it would be beneficial to provide more details on the variation in response times across different pH ranges or under different conditions.
7. Providing guidelines or recommendations for appropriate storage and handling conditions would be beneficial for users.
Reviewer 2 Report
Comments and Suggestions for Authors
In the manuscript authors Guan et al have reported responsive photonic crystal structure for PH sensing. The presented work is technically correct, which is further validated experimentally. In my opinion, this have the sufficient discussion, which can be accepted in present form with a minor correction.
Authors should compare their findings with the recent reported results at the end.
Comments on the Quality of English LanguageThere are some grammatically errors present in the manuscript. Please correct those.
Reviewer 3 Report
Comments and Suggestions for Authors
The manuscript demonstrates a photonic pH sensor based on chains of nanoparticles arrested by an agarose network. The authors clearly show the relationship of this work to their previous efforts, particularly showing that the agarose matrix relieves the need for a high magnetic field to maintain chain alignment during operation of the sensor. The work appears sound and well presented, and overall suitable for publication. Below, I identify a small number of confusing elements which may be easily improved.
All figure legends should be consistently explained in the figure captions. E.G. fig 4 may use the same numbers as other figures, but I should still see the units in the caption and what concentration (it's agarose concentration, right?).
Fig 4 looks at first glance that a and b have similar reflectance, because they are on different unlabeled unit scales. I think the comparison would be more stark if you put them on the same % scale as other plots. Then b will have a lot of white space in the top of the figure, where you could put an inset that is scaled for visibility.
"Therefore, considering both optical and mechanical properties, chain concentration of 9 mg/mL was chosen as the typical sample" I think typical is not the right word here, but I'm not sure what you were trying to say. Maybe drop this whole sentence, since the beginning of the next paragraph explains how 9 mg/mL will be important for the film thickness study.
Fig 5(d), the importance of the difference between 8.0 to 4.5 and 4.5 to 8.0 trends is not clearly explained in the text.
